# THE DEVIL IS IN THE DETAILS: ENHANCING VIDEO VIRTUAL TRY-ON VIA KEYFRAME-DRIVEN DETAILS INJECTION

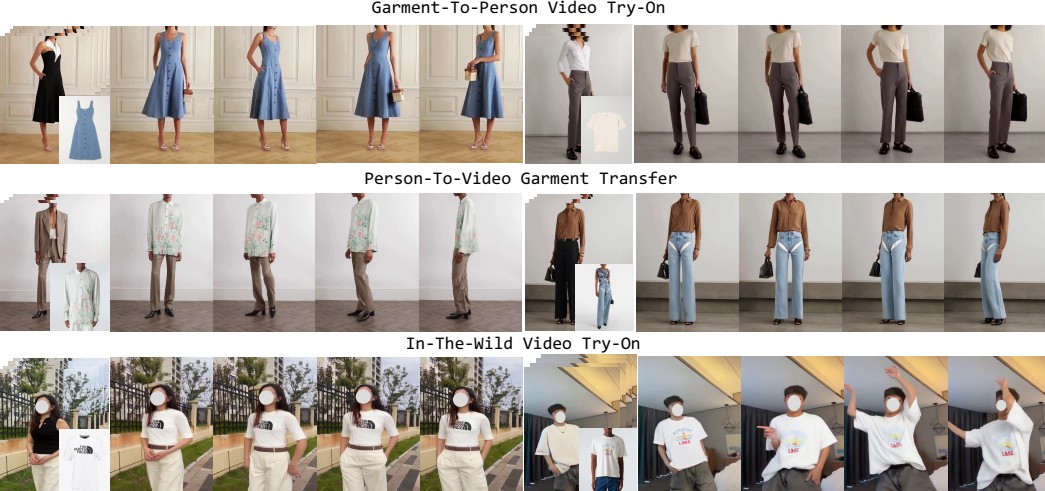

Figure 1: KeyTailor enables generating realistic and natural try-on videos with fine-grained consistency in both garment and background under challenging scenarios.

## ABSTRACT

Although diffusion transformer (DiT)-based video virtual try-on (VVT) has made significant progress in synthesizing realistic videos, existing methods still struggle to capture fine-grained garment dynamics and preserve background integrity across video frames. They also incur high computational costs due to additional interaction modules introduced into DiTs, while the limited scale and quality of existing public datasets also restrict model generalization and effective training. To address these challenges, we propose a novel framework, **KeyTailor**, along with a large-scale, high-definition dataset, **ViT-HD**. The core idea of KeyTailor is a *keyframe-driven details injection* strategy, motivated by the fact that keyframes inherently contain both foreground dynamics and background consistency. Specifically, KeyTailor adopts an instruction-guided keyframe sampling strategy to filter informative frames from the input video. Subsequently, two tailored keyframe-driven modules—the *garment details enhancement module* and the *collaborative background optimization module*—are employed to distill garment dynamics into garment-related latents and to optimize the integrity of background latents, both guided by keyframes. These enriched details are then injected into standard DiT blocks together with pose, mask, and noise latents, enabling efficient and realistic try-on video synthesis. This design ensures consistency without explicitly modifying the DiT architecture, while simultaneously avoiding additional complexity. In addition, our dataset ViT-HD comprises $15,070$ high-quality video samples at a resolution of $810 \times 1080$, covering diverse garments. Extensive experiments demonstrate that KeyTailor outperforms state-of-the-art baselines in terms of garment fidelity and background integrity across both dynamic and static scenarios. The dataset and code will be publicly released.

# 1 INTRODUCTION

The goal of video virtual try-on (VVT) is to generate natural, high-fidelity videos by substituting the clothing worn by the main character with a user-specified target garment image, while maintaining motion and visual consistency across consecutive frames. This technology not only addresses the challenge of online garment fitting for consumers on e-commerce platforms but also offers a novel and engaging experience for users on short-video platforms, making VVT an attractive direction for both industry applications and academic research.

Owing to the successful deployment of diffusion models in video generation (Blattmann et al., 2023; Wan et al., 2025; Kong et al., 2024), recent efforts in VVT increasingly employ diffusion models as their generator (Fang et al., 2024; He et al., 2024; Xu et al., 2024; Li et al., 2025b; Nguyen et al., 2025; Wang et al., 2024). These pioneers typically consist of a garment reference branch alongside a generation branch. The garment branch is responsible for extracting clothing appearance features and then interacting with the main generation branch through a tailored attention mechanism, thereby ensuring spatiotemporal consistency across frames. Although such approaches have achieved significant results, they are limited by the representational capacity of the U-Net-based (Ronneberger et al., 2015) backbone, especially when it comes to rendering complex textures and details in human motions and garment appearance (Li et al., 2025a). To overcome this limitation, recent studies (Li et al., 2025a; Chong et al., 2025b; Zuo et al., 2025) utilize large-scale video diffusion transformers (DiTs) (Peebles & Xie, 2023) in place of the U-Net backbone. This alternative not only enhances the expressiveness and scalability of the network but also enables the joint modeling of temporal and spatial patterns, thereby resulting in more consistent video generation. However, such methods still face the following challenges:

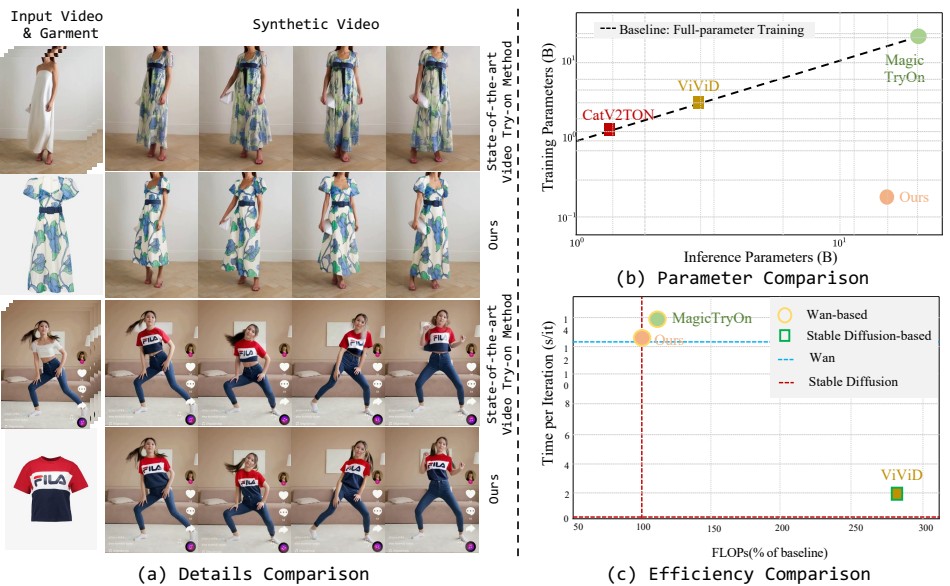

Figure 2: (a) Comparison of garment details; (b) Comparison of background details; (c) Comparison of parameters and efficienty.

*(1)* **Insufficient Garment Dynamic Details**: Although existing DiT-based methods introduce additional encoding components to learn garment appearance from both textual descriptions and visual inputs, they still fail to fully capture garment dynamic details across consecutive frames. Fine-grained cues such as backside textures, wrinkles caused by body motion (*e.g.,* raising an arm), and subtle lighting-dependent variations are often missing. As illustrated in Fig. 2(a), the SOTA method produces results with an incorrect belt position for the dress, and the generated video frames fail to capture garment variations induced by human motion (first row). In the third row, the generated garment size does not accurately correspond to the reference. These issues lead to over-smoothed garment appearances and insufficient fidelity compared to real-world dynamics.

*(2)* **Inconsistency of Background Areas**: Current methods solely rely on garment-agnostic videos to provide background conditions for video synthesis. However, this approach often results in (a) detail loss, where fine textures such as object patterns or edges are blurred; (b) temporal incon-

sistency, where elements vary unnaturally across consecutive frames, producing artifacts; and (c) environmental incoherence, where background structures deviate from the original video. For example, as shown in Fig. 2(a), the floor textures generated by the SOTA method are blurred (1st row); hair contours are inconsistent across frames, and the white frame on the wall does not align with the ground truth (3rd row). Hence, the synthesized video exhibits incoherence between garment regions and the background, and further fails to maintain background integrity, leading to degraded realism. *(3)* **Increased Model Complexity and Data Scarcity**: To enhance generation conditions, existing paradigms typically incorporate additional interaction modules into the DiT backbone. While these components improve conditioning expressiveness, they also substantially increase model complexity and computational cost. As illustrated in Fig.2(b) and Fig.2(c), we visualize the parameter counts and efficiency of SOTA methods. It is evident that SOTA methods introduce a large number of additional parameters and significantly increase training costs. In addition, currently available open-source datasets (VVT (Dong et al., 2019) and ViViD (Fang et al., 2024)) remain limited in both scale and quality. Each consists of only a few thousand short clips, often with low resolution, simple backgrounds, and limited garment diversity. These constraints prevent DiT-based methods from fully leveraging their expressive power, hindering generalization to complex scenarios and restricting high-resolution video generation.

To address the lack of fine-grained details and reduce computational cost, we propose a novel DiT-based framework, KeyTailor, built on a keyframe-driven details injection strategy. This design is motivated by the fact that informative keyframes inherently capture multi-view garment dynamics and subtle background information, which can be utilized to improve garment fidelity and background integrity. Specifically, KeyTailor employs an instruction-guided keyframe sampling approach to select frames that capture view and motion variations. This is followed by two lightweight, keyframe-driven modules for details enrichment: a garment dynamics details enhancement module, which enriches multi-view garment dynamic details (*e.g.*, wrinkles and texture variations), and a collaborative background details optimization module, which preserves structural integrity and semantic consistency in background regions. The enriched details are then fused with other conditions (*e.g.*, pose and mask latents) and injected into DiTs for realistic video synthesis. As illustrated in both Fig. 1 and Fig. 2(a), our KeyTailor ensures consistent garment dynamics and background integrity across all frames, resulting in more natural video synthesis. Importantly, KeyTailor is built upon standard DiTs without introducing additional interaction layers, and these are fine-tuned with a LoRA adapter, thereby reducing computational demand, as evidenced by Fig. 2(b) and Fig. 2(c). In addition, to combat data scarcity, we self-collect a large-scale dataset from multiple e-commerce platforms, containing 15,070 high-quality samples at a resolution of $810 \times 1080$, to facilitate training and evaluation. In summary, the contributions of this work are as follows:

- We propose KeyTailor, a novel DiT-based framework that adopts a keyframe-driven details injection strategy to enhance garment fidelity and background integrity, without introducing additional interaction layers into the DiT backbone.

- We design an instruction-guided keyframe sampling approach to select informative keyframes, along with two lightweight keyframe-driven details injection modules–a garment dynamic details enhancement module and a collaborative background details optimization module–to ensure that the details injected into DiTs provide sufficient garment fidelity and background integrity.

- We curate ViT-HD, a new large-scale and high-definition dataset collected from multiple e-commerce platforms, containing $15,070$ video samples at a resolution of $810 \times 1080$, across various garment styles.

- We conduct extensive experiments on ViT-HD, as well as on two widely used VVT datasets and two image-based virtual try-on datasets, demonstrating that KeyTailor outperforms state-of-the-art baselines in both garment fidelity and background consistency across dynamic and static scenarios.

## 2 VIT-HD DATASET

Existing public datasets, VVT (Dong et al., 2019) and ViViD (Fang et al., 2024), either suffer from extremely low resolution–resulting in the loss of fine-grained garment textures and details–or are restricted to simple, repetitive runway scenes. Moreover, their limited scale still falls short of the growing need for large, high-quality video data. To address these limitations, we curate a new dataset, ViT-HD, which significantly expands both the scale and quality of available resources. Table 1 pro-

Table 1: Dataset Comparison. We compare ViT-HD with existing video virtual try-on datasets along four dimensions: resolution, garment diversity (multi-class), video content quality (no start-frame overexposure and intact subject integrity), and data scale.

| Datasets | Resulution | Multi Class | No Start Overexposure | Subject Integrity | Scale |
|---|---|---|---|---|---|
| VVT | $192 \times 256$ | × | ✓ | × | 791 |
| ViViD | $632 \times 824$ | ✓ | × | × | 9,700 |
| ViT-HD (Ours) | $810 \times 1080$ | ✓ | ✓ | ✓ | 15,070 |

vides a detailed comparison between our dataset and existing ones. Our proposed ViT-HD contains $15,070$ samples featuring diverse garment styles, each with a resolution of $810 \times 1080$.

**Data collection and processing**: Our raw data are downloaded from multiple e-commerce platforms. Each raw data sample consists of a high-resolution garment image together with a corresponding high-definition model show-case video, both cropped at a resolution of $1080 \times 810$. Then, for each raw data sample, we follow ViViD (Fang et al., 2024) to extract its pose video and masked video. Specifically, we employ OpenPose (Cao et al., 2019) to detect skeletal keypoints and generate pose sequences. To obtain masked videos, we adopt the same segmentation pipeline as OOTDiffu-

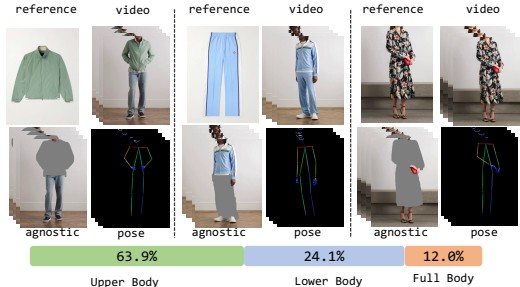

Figure 3: Dataset overview.

sion (Xu et al., 2025). For each frame, we utilize OpenPose (Cao et al., 2019) together with HumanParsing (Li et al., 2020) to infer body-part masks and create a garment-agnostic background image by inpainting the clothed regions. The resulting frames are then post-processed and stitched together to form the masked video. Furthermore, we categorize each data sample into one of three types–upper-body, lower-body, or full-body outfits–using BLIP-2 (Li et al., 2023). During the data processing stage, we discard videos that contain a large number of frames with incomplete clothing occlusion to preserve subject integrity. In addition, we remove overexposed frames at the beginning of the original videos to maintain consistent color tones across all frames within each video. Fig. 3 presents an overview of ViT-HD.

## 3 METHOD

Our KeyTailor is built upon diffusion transformers (DiTs), aiming to provide a lightweight solution for synthesizing realistic, high-fidelity videos that capture dynamic garment details while maintaining background consistency. To this end, we propose a keyframe-driven details injection strategy with two tailored feature extraction modules: a *garment dynamics enhancement module* and a *collaborative background optimization module*. Specifically, these two modules strengthen garment dynamics and background integrity by leveraging information from keyframes as supplementary input. The enriched fine-grained details are then directly injected into DiTs to introduce multi-view garment variations and preserve background consistency—without explicitly introducing interaction modules into the DiT architecture, as required in prior work (Chong et al., 2025b; Li et al., 2025a; Zuo et al., 2025). The overall framework of KeyTailor is illustrated in Fig. 4.

### 3.1 KEYFRAME-DRIVEN DETAILS INJECTION

The goal of VVT is to synthesize a new video by replacing the clothing worn by a character, while preserving all other aspects—such as motion, background, and garment dynamics—consistent with the original video. We argue that keyframes naturally capture critical information about both garment variations and background consistency, making them highly beneficial for guiding DiTs. Based on this assumption, we propose a keyframe-driven details injection strategy. This strategy is implemented through an instruction-guided keyframe sampling module and two lightweight keyframe-driven injection modules: a garment dynamic details enhancement module and a collaborative background details optimization module.

**Instruction-guided keyframe sampling**. Effectively selecting informative keyframes is the key to our keyframe-driven details injection. To ensure that the selected frames adequately capture both view changes (*e.g.,* front and back) and action changes (*e.g.,* raising a hand) from $V_{in}$, we propose an instruction-guided keyframe sampling module (IKS). IKS first employs a large visual-

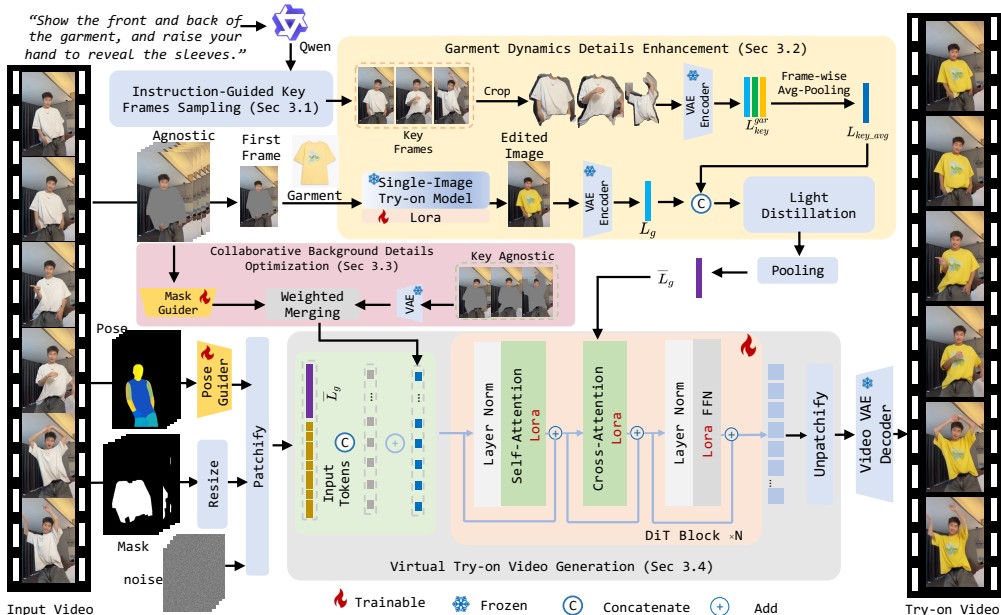

Figure 4: **Overall framework of KeyTailor.** KeyTailor takes as input a reference garment image $I_{ref}$, a source video $V_{in}$, its corresponding agnostic video $V_{agn}$, agnostic masks $M_{agn}$, and pose representations $P$. These inputs are encoded into garment-related latents $L_g$, background-related latents $L_{bg}$, pose latents $L_p$, and resized masks $L_m$. Specifically, garment-related latents are generated by the GDDE module, background-related latents by the CBDO module, and pose latents by a trainable pose guider. Subsequently, all these latents, together with noise latents, are injected into $N$ DiT blocks to produce the final try-on video tokens, which are then decoded by a VAE-based video decoder to synthesize the output video.

language model (*e.g.,* QWen (Bai et al., 2023)) to parse the predefined view–action instruction and extract the target views $\mathcal{V}_{tar}$ and actions $\mathcal{A}_{tar}$, which reflect view variation and action dynamics, respectively. Then, IKS applys HumanParsing (Li et al., 2020) to generate standardized multi-anchor pose frames $F_{anc}$ corresponding to $\mathcal{V}_{tar}$ and $\mathcal{A}_{tar}$. Subsequently, for each frame $f \in V_{in}$, IKS computes a motion-difference score $S_m(f)$ with respect to $F_{anc}$, along with a garment-area ratio score $S_r(f)$ (see Appendix E for details). The final score of the frame $f$ is computed as:

$$S_f(f) = 1 - S_m(f) + \lambda \cdot S_r(f), \tag{1}$$

where $\lambda$ is a balancing coefficient. Next, all frames are sorted in descending order according to their $S_f$. Rather than simply selecting the top-$k$ frames to construct the final multi-view keyframes $F_{key}$, we adopt a dual-selection strategy to reduce redundancy and ensure temporal uniformity. Specifically, two thresholds are defined to constrain both the score difference and the temporal interval between the current frame $f$ and candidate frames $f_{key} \in F_{key}$. The pseudo-code for this procedure is provided in Algorithm 1 in Appendix H.

**Garment Dynamic Details Enhancement**. Unlike previous works that learn garment appearance solely by embedding the garment image $I_{ref}$ or incorporating textual descriptions, our garment dynamic details enhancement module (GDDE) instead encodes the edited first-frame result and further enriches it with features extracted from keyframes. Specifically, GDDE first crops the initial frame $f_{agn}^0$ from the agnostic video $V_{agn}$, and employs a pre-trained single-image try-on model with LoRA (Hu et al., 2022) layers to inject the garment appearance into $f_{agn}^0$. The resulting try-on frame is then projected into a latent representation $L_g$ using a pre-trained VAE-based image encoder $\mathcal{E}_{VAE}$. Then, GDDE enriches $L_g$ with fine-grained garment variations derived from $F_{key}$, such as backside textures and wrinkles caused by raised arms. It first extracts garment-specific features $L_{key}^{gar}$ from $F_{key}$ by encoding the garment regions of each keyframe using $\mathcal{E}_{VAE}$, where the garment regions are obtained through the segmentation operation. Subsequently, GDDE employs a lightweight distillation

component $\mathcal{D}$ to inject garment variation details from $L_{key}^{gar}$ into $L_g$, formulated as:

$$\bar{L}_g = \mathcal{D}(\text{Concat}(L_g, \frac{1}{|F_{key}|} \sum_{L_k \in L_{key}^{gar}} L_k)). \quad (2)$$

Here, $\mathcal{D}$ is implemented using two $1 \times 1$ convolution layers followed by a LayerNorm layer.

**Collaborative Background Details Optimization**. Preserving background integrity is crucial for synthesizing realistic video scenes. Existing methods typically encode the garment-agnostic video $V_{agn}$ into a latent representation as the background condition for DiTs. However, since $V_{agn}$ is generated by applying image inpainting to each frame of the original video to fill in garment regions, it inevitably loses subtle background details. To this end, we design a collaborative background details optimization module (CBDO), which introduces keyframes as supplementary cues to enrich the semantics of $V_{agn}$. Specifically, CBDO consists of two branches: a *coarse global background encoding branch* and a *fine-grained keyframe-driven local detail enrichment branch*. In the first branch, CBDO employs a mask guider $\mathcal{E}_{BG}$ to project $V_{agn}$ into a latent representation $L_{bg}$, thereby capturing the global structural layout and semantic context of the background. $\mathcal{E}_{BG}$ is implemented with four 3D convolutional layers, having channel dimensions of 32, 96, 192, and 256, respectively. To facilitate smoother latent guidance during video synthesis training, the linear layer in $\mathcal{E}_{BG}$ is zero-initialized. In the second branch, CBDO extracts subtle background details from keyframes. Specifically, it first crops the background regions using an inverse human-body mask operation and then encodes them into latent representations with $\mathcal{E}_{VAE}$, denoted as $L_{key}^{bg}$. To avoid redundancy and ambiguity in fine-grained background details, we select only the frame with the highest background completeness score as the supplementary input to $L_{bg}$, yielding the enhanced background latents:

$$\bar{L}_{bg} = \alpha \cdot L_{bg} + (1 - \alpha)L_{key}^{max}, \quad (3)$$

where $\alpha$ is a balance weight, with a default setting to 0.3.

## 3.2 VIRTUAL TRY-ON VIDEO GENERATION

After obtaining the enhanced garment-related latents $\bar{L}_g$ and optimized background-related latents $\bar{L}_{bg}$, we adopt a *three-step fusion strategy* rather than directly concatenating them with the pose latent $L_p$, the resized agnostic masks $L_m$, and the noise $\epsilon$ as input to the DiTs. First, $L_p$ and $L_m$ are concatenated and patchified into input tokens $T_{inp}$, which are then fused with $\bar{L}_g$ through a projection layer $\mathcal{R}$ to produce $L$. $L$ is then concatenated with patchified $\epsilon$ to form $\bar{L}$. Finally, $\bar{L}_{bg}$ is injected into $\bar{L}$ via the "addto" operation, yielding the final guidance tokens for DiTs. In this way, the guidance preserves fine-grained background details while maintaining pose structure and garment dynamics. During the denoising step, we stack $N$ DiT blocks and apply LoRA to finetune their attention modules, including both self-attention and cross-attention. Moreover, the garment-related latents $\bar{L}_g$ is injected into the cross-attention component, substituting the original text tokens to mitigate the loss of garment details. Architecturally, KeyTailor only performs detail injection without modifying any component of the original DiT architecture, thereby avoiding the introduction of massive training parameters compared to prior works (Li et al., 2025a). After several denoising iterations within the DiT backbone, the network generates try-on video tokens, which are subsequently decoded into video sequences by the Video VAE decoder. The training details of KeyTailor are described in Appendix F.

## 4 EXPERIMENTS

### 4.1 SETUPS: DATASETS, METRICS AND DETAILS

**Datasets**: We conduct experiments on our proposed dataset ViT-HD, as well as on the publicly available VVT (Dong et al., 2019) and ViViD (Fang et al., 2024) datasets. ViViD contains 7,759 paired video training samples, while ViT-HD provides 13,070 paired training samples and 2,000 test samples after partitioning. For training, we combine ViT-HD with ViViD, and for testing, we additionally evaluate on indoor scenarios using the ViViD-S and VVT datasets. To further assess generalization ability, we also perform experiments on two image-based virtual try-on datasets, VITON-HD (Choi et al., 2021) and DressCode (Morelli et al., 2022).

**Metrics**: Following previous works (Zhang et al., 2018; Wang et al., 2004; Carreira & Zisserman, 2017; Hara et al., 2018), we adopt three widely used metrics to evaluate video generation quality: SSIM, LPIPS, and VFID. SSIM measures the structural similarity between generated and

reference videos, LPIPS captures perceptual differences between image pairs, and VFID assesses both temporal consistency and overall video quality, where I3D (Carreira & Zisserman, 2017) and ResNeXt (Xie et al., 2017) are different backbone models. For image virtual try-on, we additionally use FID and KID, along with SSIM and LPIPS (Li et al., 2025a). Herein, FID and KID measure the similarity between the distributions of two images.

**Implementation Details**: We adopt the pre-trained weights from Wan2.1-I2V-14B-720P as the base model. To address overexposure and subject loss in the opening frames of ViViD videos, we truncate the initial frames of each sequence. During training, each video sample consists of 81 frames, with a batch size of 1, and training is performed for 14,500 iterations. We use the AdamW optimizer with a fixed learning rate of 1e-4. FiTDiT (Jiang et al., 2024) is employed as the image-based virtual try-on model, following the official default settings. For video inference, the number of inference steps is set to 25. To ensure fairness, all model variants in the ablation studies are evaluated under the same hyperparameter configurations during inference.

Table 2: Quantitative Comparison of Video Virtual Try-On Results on ViT-HD. The best and second-best results are marked with red and blue, respectively. $p$ and $u$ denote the paired setting and unpaired setting, respectively.

| Methods | Venue | $VFID_I^p\downarrow$ | $VFID_R^p\downarrow$ | SSIM↑ | LPIPS↓ | $VFID_I^u\downarrow$ | $VFID_R^u\downarrow$ |
|---|---|---|---|---|---|---|---|
| *Image-centered Method* | | | | | | | |
| StableVITON (Kim et al., 2024) | CVPR'24 | 38.2686 | 0.8021 | 0.7986 | 0.1608 | 39.2286 | 0.8525 |
| OOTDiffusion (Xu et al., 2025) | AAAI'25 | 30.2521 | 4.0068 | 0.7925 | 0.1125 | 38.5214 | 5.5221 |
| CatVTON (Chong et al., 2025a) | ICLR'25 | 22.2365 | 0.4028 | 0.8156 | 0.1325 | 28.2065 | 0.7352 |
| *Video-centered Method* | | | | | | | |
| ViViD (Fang et al., 2024) | arxiv'24 | 19.0568 | 0.7525 | 0.8022 | 0.1363 | 22.6856 | 0.7925 |
| CatV2TON (Chong et al., 2025b) | CVPRW'25 | 15.8725 | 0.2898 | 0.8545 | 0.0976 | 20.0187 | 0.5762 |
| MagicTryOn Li et al. (2025a) | arxiv'25 | 14.0587 | 0.2461 | 0.8622 | 0.0828 | 19.2253 | 0.5587 |
| Ours | Ours | 7.5267 | 0.1628 | 0.9066 | 0.0397 | 13.6628 | 0.3519 |

Table 3: Quantitative Comparison of Video Virtual Try-On Results on VVT (Dong et al., 2019) (***Left***) and ViViD (Fang et al., 2024) (***Right***).

| Methods | $VFID_I^p\downarrow$ | $VFID_R^p\downarrow$ | SSIM↑ | LPIPS↓ |
|---|---|---|---|---|
| FW-GAN | 8.019 | 0.1215 | 0.675 | 0.283 |
| MV-TON | 8.367 | 0.0972 | 0.853 | 0.233 |
| ClothFormer | 3.967 | 0.0505 | 0.921 | 0.081 |
| ViViD | 3.793 | 0.0348 | 0.822 | 0.107 |
| CatV2TON | 1.778 | 0.0103 | 0.900 | 0.039 |
| MagicTryOn | 1.991 | 0.0084 | 0.958 | 0.024 |
| Ours | 1.226 | 0.0059 | 0.968 | 0.016 |

| Methods | $VFID_I^p\downarrow$ | $VFID_R^p\downarrow$ | SSIM↑ | LPIPS↓ | $VFID_I^u\downarrow$ | $VFID_R^u\downarrow$ |
|---|---|---|---|---|---|---|
| StableVITON | 34.2446 | 0.7735 | 0.8019 | 0.1338 | 36.8985 | 0.9064 |
| OOTDiffusion | 29.5253 | 3.9372 | 0.8087 | 0.1232 | 35.3170 | 5.7078 |
| IDM-VTON | 20.0812 | 0.3674 | 0.8227 | 0.1163 | 25.4972 | 0.7167 |
| ViViD | 17.2924 | 0.6209 | 0.8029 | 0.1221 | 21.8032 | 0.8212 |
| CatV2TON | 13.5962 | 0.2963 | 0.8727 | 0.0639 | 19.5131 | 0.5283 |
| MagicTryOn | 12.1988 | 0.2346 | 0.8841 | 0.0815 | 17.5710 | 0.5073 |
| DreamVVT | 11.0180 | 0.2549 | 0.8737 | 0.0619 | 16.9468 | 0.4285 |
| Ours | 8.2164 | 0.1854 | 0.8768 | 0.0522 | 14.1521 | 0.3866 |

Table 4: Quantitative Comparison of Image Virtual Try-on Results on VITON-HD (Choi et al., 2021) and DressCode (Morelli et al., 2022).

| | Metric | Methods | | | | | | | |
|---|---|---|---|---|---|---|---|---|---|
| | | GP-VTON | LaDI-VTON | IDM-VTON | OOTDiffusion | CatVTON | CatV2TON | MagicTryOn | Ours |
| VITON-HD | $FID^p\downarrow$ | 8.726 | 11.386 | 6.338 | 9.305 | 6.139 | 8.095 | 8.036 | 5.293 |
| | $KID^p\downarrow$ | 3.944 | 7.248 | 1.322 | 4.086 | 0.964 | 2.245 | 1.235 | 0.720 |
| | SSIM↑ | 0.8701 | 0.8603 | 0.8806 | 0.8187 | 0.8691 | 0.8902 | 0.8936 | 0.9201 |
| | LPIPS↓ | 0.0585 | 0.0733 | 0.0789 | 0.0876 | 0.0973 | 0.0572 | 0.0477 | 0.0566 |
| | $FID^u\downarrow$ | 11.844 | 14.648 | 9.611 | 12.408 | 9.143 | 11.222 | 8.696 | 8.528 |
| | $KID^u\downarrow$ | 4.310 | 8.754 | 1.639 | 4.689 | 1.267 | 2.986 | 1.130 | 0.788 |
| DressCode | $FID^p\downarrow$ | 9.927 | 9.555 | 6.821 | 4.610 | 3.992 | 5.722 | 5.428 | 2.746 |
| | $KID^p\downarrow$ | 4.610 | 4.683 | 2.924 | 0.955 | 0.818 | 2.338 | 1.078 | 0.517 |
| | SSIM↑ | 0.7711 | 0.7656 | 0.8797 | 0.8854 | 0.8922 | 0.9222 | 0.9572 | 0.9621 |
| | LPIPS↓ | 0.1801 | 0.2366 | 0.0563 | 0.0533 | 0.0455 | 0.0367 | 0.0271 | 0.0347 |
| | $FID^u\downarrow$ | 12.791 | 10.676 | 9.546 | 12.567 | 6.137 | 8.627 | 6.962 | 5.147 |
| | $KID^u\downarrow$ | 6.627 | 5.787 | 4.320 | 6.627 | 1.549 | 3.838 | 0.908 | 1.012 |

## 4.2 PERFORMANCE COMPARISON WITH SOTA METHODS

**Quantitative Comparison**: Table 2 and Table 3 report the results of SOTA baselines and our Key-Tailor for the video virtual try-on task on ViT-HD, VVT, and ViViD, respectively. It is evident that KeyTailor outperforms existing SOTA methods across nearly all metrics in both paired and unpaired settings. This demonstrates that KeyTailor achieves superior visual quality and temporal consistency in synthesized videos. We attribute this improvement to the injection of keyframe information, which provides both garment dynamics and subtle background details. Leveraging these cues enhances garment fidelity and preserves background integrity consistently across video frames. Additionally, the results on the image virtual try-on task (Table 4), show that KeyTailor also delivers better performance in static scenarios, further validating its generalization ability.

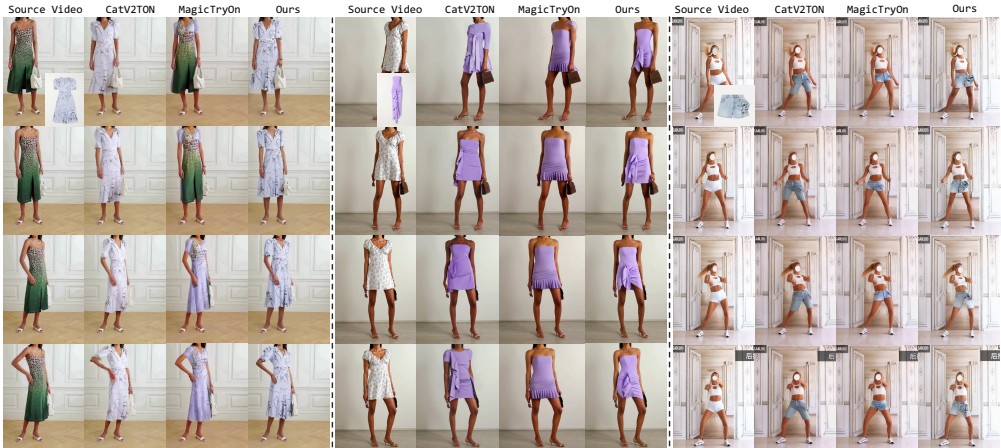

Figure 5: Qualitative comparison of video virtual try-on results on the ViViD dataset (1st column), our ViT-HD dataset (2nd column), and in-the-wild scenarios (3rd column). Our KeyTailor restores fine-grained garment details while preserving background integrity.

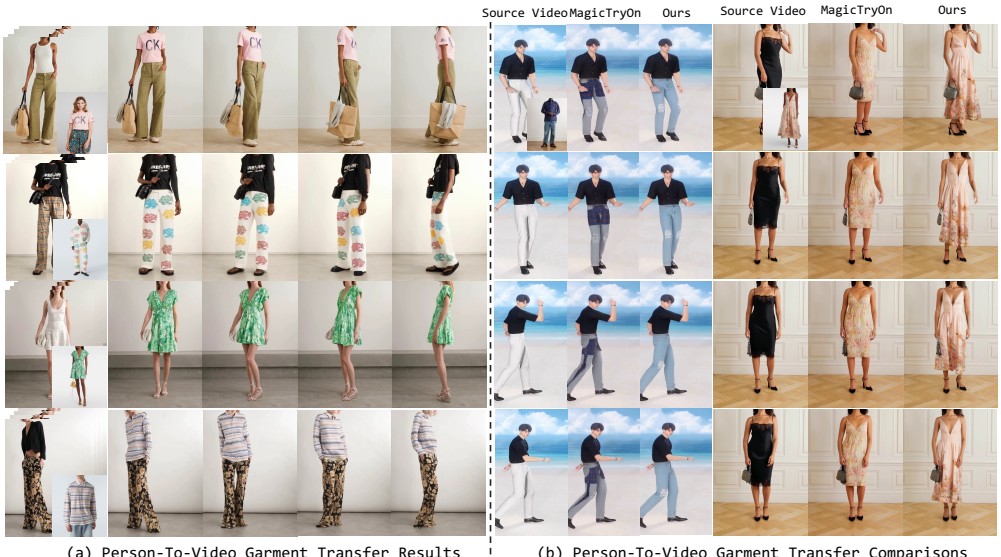

(a) Person-To-Video Garment Transfer Results     (b) Person-To-Video Garment Transfer Comparisons

Figure 6: Qualitative results and comparisons in person-to-video garment transfer scenarios. Our method combines background, person, and garment more naturally in complex scenarios.

**Qualitative Comparison**: We present the synthesized results of baselines and our method in Fig. 5 and Fig. 6. In Fig. 5, we visualize results on ViViD and our dataset, as well as an example under an in-the-wild scenario. It can be observed that KeyTailor not only preserves garment details but also maintains background consistency. Specifically, our method retains more garment details and patterns, adapts better to human motion, and produces more reasonable and natural fits, whereas

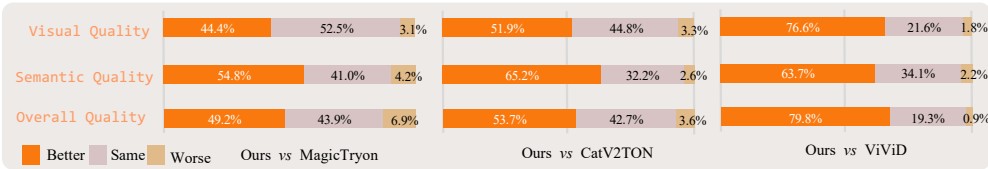

Figure 7: User Study. We report pairwise preference rates from the perspectives of visual quality, semantic consistency, and overall quality.

other methods often lose details or even alter garment styles. Additionally, regions outside the garment remain more consistent with the original video in our results, exhibiting fewer artifacts and clearer background structures. In Fig. 6, we evaluate SOTA methods and our KeyTailor using a more challenging reference garment image, *i.e.,* person-to-video garment transfer. Our KeyTailor still preserves fine-grained garment details and maintains background consistency, demonstrating robust performance even under complex garment transfer scenarios.

**User Study.** We conduct a user study following the standard win-rate methodology to evaluate our approach. Each questionnaire contains 15 randomly selected generated videos presented in randomized order, and participants are asked to evaluate them along three dimensions: visual quality, semantic consistency, and overall quality. In total, we collect 80 completed feedback forms, with the results presented in Fig. 7. The majority of participants prefer our KeyTailor over SOTA methods, with particularly clear advantages compared to CatV2TON and ViViD. These findings demonstrate that our method produces more realistic, coherent, and user-preferred video try-on results.

## 4.3 ABLATION STUDY

We conduct an ablation study by deactivating individual modules (see Appendix G for details) to demonstrate the effectiveness of each component in KeyTailor. The results are reported in Table 5. Overall, the full version of KeyTailor outperforms all variants. Removing any component leads to a clear performance degradation, with the garment dynamics distillation (*w/o* $\mathcal{D}$) showing the most significant drop. The results of "*w/o* IKS" and "$F_{key} = 1$" demonstrate that the IKS module can accurately select informative frames and highlight the importance of using multiple keyframes. Fig. 8 provides an intuitive visual comparison, show-

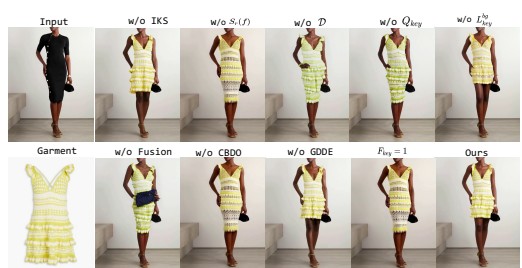

Figure 8: Qualitative comparison of of KeyTailor with variants on ViT-HD.

ing that removing these components results in noticeable generation errors, such as unnatural color shifts, distorted backgrounds, hallucinated objects, and skin tone changes.

Table 5: Ablation study of each component on the ViT-HD dataset.

| Metric | *w/o* IKS | *w/o* $S_r(f)$ | *w/o* $\mathcal{D}$ | *w/o* $Q_{key}$ | *w/o* $L_{key}^{bg}$ | *w/o* Fusion | *w/o* CBDO | *w/o* GDDE | $F_{key} = 1$ | Ours |
|---|---|---|---|---|---|---|---|---|---|---|
| VFID$_I^p\downarrow$ | 16.2586 | 17.2568 | 22.5241 | 15.9878 | 16.2526 | 16.2885 | 17.2134 | 19.8968 | 16.3856 | 7.5267 |
| VFID$_R^p\downarrow$ | 0.3568 | 0.4566 | 0.5869 | 0.8201 | 0.6645 | 0.6022 | 0.7826 | 0.5863 | 0.3988 | 0.1628 |
| SSIM$\uparrow$ | 0.8035 | 0.8461 | 0.7658 | 0.8569 | 0.8622 | 0.8254 | 0.8523 | 0.8429 | 0.8165 | 0.9066 |
| LPIPS$\downarrow$ | 0.1022 | 0.1125 | 0.2106 | 0.0807 | 0.0823 | 0.0935 | 0.0982 | 0.1136 | 0.0987 | 0.0397 |
| VFID$_I^u\downarrow$ | 21.6326 | 22.0187 | 25.3632 | 21.5855 | 22.3523 | 23.5525 | 22.3965 | 23.5662 | 21.8867 | 13.6628 |
| VFID$_R^u\downarrow$ | 0.5679 | 0.5602 | 0.7901 | 0.6988 | 0.8725 | 0.8263 | 0.6885 | 0.8255 | 0.5969 | 0.3519 |

## 5 CONCLUSION

In this work, we present KeyTailor, a novel DiT-based video virtual try-on model, along with ViT-HD, a large-scale high-definition dataset. KeyTailor is designed based on a keyframe-driven details injection strategy, implemented via an instruction-guided keyframe sampling module and two lightweight keyframe-driven details injection modules: the garment dynamic details enhancement and the collaborative background details optimization. Extensive experiments on both video and image try-on tasks show that KeyTailor achieves superior garment fidelity, background integrity, and temporal consistency, establishing strong baselines and resources for future research in this field.

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

APPENDIX

## A  REPRODUCIBILITY STATEMENT

We have already elaborated on all the models or algorithms proposed, experimental configurations, and benchmarks used in the experiments in the main body or appendix of this paper. Furthermore, we declare that the entire code used in this work will be released after acceptance.

## B  THE USE OF LARGE LANGUAGE MODELS

We use large language models solely for polishing our writing, and we have conducted a careful check, taking full responsibility for all content in this work.

## C  RELATED WORK

Video virtual try-on (VVT) aims to replace a person's clothing with a target garment while preserving the spatiotemporal consistency of the video, i.e., the generated results should ensure a consistent appearance of the target garment across frames, align seamlessly with the person's pose and motion, and maintain the rest of the scene without distortion.

**GAN-based methods**. Earlier works adopt GAN-based generators for VVT (Dong et al., 2019; Choi et al., 2021; Zhong et al., 2021; Jiang et al., 2022). For instance, FW-GAN (Dong et al., 2019), as the first attempt, introduces an optical flow–guided warping GAN to deform the target garment and align it with the character's body, thereby generating coherent video frames. MV-TON (Zhong et al., 2021) leverages memory refinement to enhance details from previously generated frames. ClothFormer (Jiang et al., 2022) employs a dual-stream transformer to address the temporal consistency of warped input sequences.

**Diffusion model-based methods**. Inspired by the advances of diffusion models in video generation (Blattmann et al., 2023; Wan et al., 2025; Kong et al., 2024), the latest studies (Fang et al., 2024; Chong et al., 2025b; Xu et al., 2024; Zheng et al., 2024) have delved deeper into developing diffusion model–based frameworks for solving VVT tasks. For instance, ViVid (Fang et al., 2024) and Tunnel Try-on (Xu et al., 2024) incorporate a clothing reference branch into the stable diffusion framework to inject appearance features of the target garment. They further introduce temporal modeling strategies to ensure temporal coherence across frames, achieving visually plausible and temporally consistent video generation. However, these pioneering methods treat spatial and temporal information separately and are limited to low-quality outputs due to the restricted expressive capacity of U-Net (Ronneberger et al., 2015) backbones.

**DiT-based methods**. To achieve superior realism and fine-grained detail preservation in virtual try-on applications, subsequent efforts (Chong et al., 2025b; Li et al., 2025a; Zuo et al., 2025) have explored diffusion transformer (DiT)–based diffusion models. These methods typically adopt a dual-branch architecture, where a dedicated clothing branch encodes appearance features, which are then combined with other input conditions and fed into DiT blocks for garment transfer. The DiT structure facilitates the joint modeling of spatial and temporal consistency. Furthermore, additional interaction modules are incorporated into the DiT blocks to better preserve garment details.

Despite the significant advancements of these DiT-based methods, they still suffer from limitations in fully capturing garment dynamics and background details. Furthermore, the introduction of additional components into DiT blocks often leads to increased complexity and computational overhead. Finally, the broader applicability of the DiT architecture remains constrained by the limited scale of publicly available datasets for virtual try-on tasks. Our KeyTailor offers a more lightweight solution that improves garment fidelity and background consistency by injecting keyframe-driven details without explicitly modifying the DiT architecture. In addition, we curate a large-scale, high-quality dataset ViT-HD, to address the issue of data scarcity.

## D  PRELIMINARY

**Diffusion Transformers** (DiTs) have revolutionized video generation with their strong expressive power, ensuring higher fidelity and alignment in video synthesis. Wan (Wan et al., 2025) represents the current state-of-the-art among open-source video diffusion models, offering broad practical value. It adopts an LLaVA-style architecture (Liu et al., 2023), consisting of a variational autoencoder (VAE), a text encoder umT5 (Chung et al., 2023), and a DiT backbone. The core DiT network is composed of a patchifying module, multiple Transformer blocks, and an unpatchifying module. To ensure effective instruction following in long-context scenarios, Wan alternates between cross-attention and self-attention mechanisms (Chen et al., 2021; Zhang et al., 2019).

Wan processes noisy video tokens $\mathbf{x}0 \in \mathbb{R}^{N \times d}$ and text condition embeddings $c\text{txt} \in \mathbb{R}^{M \times d}$, where $\mathbf{x}_0 \sim \mathcal{N}(0, I)$, $d$ denotes the embedding dimension, and $N$ and $M$ represent the number of video and text tokens, respectively. Leveraging flow matching, Wan circumvents iterative velocity prediction. In flow matching, given a latent representation of the target video $x_1$, a random noise sample $x_0$, and a timestep $t \in [0, 1]$ sampled from a logit-normal distribution, the linear interpolation $x_t$ between $x_0$ and $x_1$ is used as the training input:

$$x_t = tx_1 + (1 - t)x_0. \tag{A1}$$

The corresponding ground truth velocity field is:

$$v_t = \frac{dx_t}{dt} = x_1 - x_0. \tag{A2}$$

**Low-Rank Adaptation** (LoRA) (Hu et al., 2022) is a parameter-efficient finetuning technique that introduces low-rank matrices to adapt pretrained models without directly updating the original weights. Given a pre-trained weight matrix $W_0 \in \mathbb{R}^{d \times k}$, LoRA approximates the updated weight as:

$$W = W_0 + \Delta W = W_0 + AB^T, \tag{A3}$$

where $A \in \mathbb{R}^{d \times r}$ and $B \in \mathbb{R}^{k \times r}$ are trainable low-rank matrices with rank $r \ll \min(d, k)$. This low-rank decomposition significantly reduces the number of trainable parameters, enabling efficient adaptation while preserving the performance of the original model.

## E  SCORE DEFINITION

In this section, we provide the definitions of the score functions used in our work, i.e., motion difference score ($S_m(f)$), garment-area ratio score ($S_r(f)$), and background integrity score ($S_{bg}(f)$). The motion difference score is used to quantify the discrepancy between the motion of a given frame and that of the anchor frames, and is defined as:

$$S_m(f) = \min_{f_a \in F_{anc}} \big( \cos(D(f), D(f_a)) \big), \tag{A4}$$

where $\cos(\cdot, \cdot) \in [0, 1]$. A lower value indicates a larger difference in skeleton direction.
The garment-area ratio score measures the proportion of a frame occupied by the garment, and its formulation is given as:

$$S_r(f) = \frac{\text{area}(\text{segment-cloth}(f))}{\text{area}(f)}. \tag{A5}$$

Here, "segment-cloth" denotes the segmentation operation that extracts the garment area from the frame, implemented by using HumanParsing (Li et al., 2020).
Similarly, the background integrity score measures the clarity and proportion of the preserved background, and is calculated as:

$$S_{bg}(f) = \text{Background Ratio}(f) \times \text{Clarity}(f), \tag{A6}$$

where the background ratio is:

$$\text{Background Ratio}(f) = \frac{\text{area}(\text{segment-background}(f))}{\text{area}(f)}. \tag{A7}$$

Specifically, the segment-background$(f)$ is accomplished via HumanParsing (Li et al., 2020), which first extracts the human region, and the background is then computed as

$$\text{segment-background}(f) = f \odot \big(1 - \text{segment-human}(f)\big). \tag{A8}$$

And the clarity for frame $f$ is computed as:

$$\text{Clarity}(f) = \left( \frac{|\{E(x, y) > T\}|}{|\{\text{background pixels}\}|} \right) \times \left( \frac{1}{255} \cdot \frac{\sum_{E(x,y)>T} E(x, y)}{|\{E(x, y) > T\}|} \right), \tag{A9}$$

where $E$ is the Sobel edge map of the background image, $T$ is a fixed threshold (default 50). The first term denotes the edge density, and the second term represents the normalized average gradient strength.

## F    TRAINING DETAILS

As shown in Fig. 4, finetuning is applied to the LoRA parameters added to the DiT blocks, the single-image try-on model, and the parameters of the mask guider and pose guider. All of these are initialized from pre-trained weights, while the remaining modules are kept frozen. The formulations of the LoRA weights for finetuning image virtual try-on are shown as follows:

$$
\begin{aligned}
Q_{Img} &= W_{Q,I} + A_{Q,I} B_{Q,I}^{\top}, \\
K_{Img} &= W_{K,I} + A_{K,I} B_{K,I}^{\top}, \\
V_{Img} &= W_{V,I} + A_{V,I} B_{V,I}^{\top},
\end{aligned}
\tag{A10}
$$

Similarly, the LoRA weights for finetuning DiT in our KeyTailor are defined as:

$$
\begin{aligned}
Q_{DiT} &= W_{Q,D} + A_{Q,D} B_{Q,D}^{\top}, \\
Q_{key} &= Q_{DiT} + A_{key} B_{key}^{\top} \cdot L_{avg\text{-}key}^{\top}, \\
K_{DiT} &= W_{K,D} + A_{K,D} B_{K,D}^{\top}, \\
V_{DiT} &= W_{V,D} + A_{V,D} B_{V,D}^{\top},
\end{aligned}
\tag{A11}
$$

where $W_*$ denotes the pre-trained weight matrix of the original projection layers, and $A_*$ represents a low-rank trainable matrix with rank $r \ll \min(d,k)$. $Q_{key}$ corresponds to the keyframe. This parameter-efficient adaptation enables dynamic modulation of attention mechanisms while preserving the original model capacity. We further apply LoRA to the linear transformations in the feed-forward network (FFN), where each weight matrix $W_{F0}$ is similarly parameterized as:

$$
W_F = W_F + A_F B_F^{\top}.
\tag{A12}
$$

This extension allows for lightweight finetuning across both attention and MLP components of the diffusion transformer.

By leveraging flow matching to maintain equivalence with the maximum likelihood objective, the model is trained to learn the true velocity. The overall training objective $\mathcal{L}$ is defined as:

$$
\mathcal{L} = \mathbb{E}_{c_{\text{txt}},t,x_1,x_0} \left[ \| u(x_t, L, c_{\text{txt}}, t; \theta) - v_t \|_2^2 \right]
\tag{A13}
$$

where $u(x_t, L, c_{\text{txt}}, t; \theta)$ denotes the velocity predicted by the model.

## G    SETTINGS OF ABLATION STUDY

In this section, we present the detailed settings of the ablation study in our experiments.

- **w/o IKS**: Replace instruction-guided keyframe sampling with random sampling of three frames from the input video. This variant validates the effectiveness of the instruction-guided keyframe sampling module.

- **w/o $S_r(f)$**: Use only the motion-difference score $S_m(f)$ to guide keyframe selection, without incorporating user-instruction parsing, keeping the rest unchanged. This variant evaluates the contribution of instruction guidance.

- **w/o $\mathcal{D}$**: Directly use the first frame of the input video to generate garment latents, without incorporating keyframe-driven garment details, keeping the rest unchanged. This variant examines the role of the distillation component.

- **w/o $Q_{key}$**: Continue injecting $\bar{L}_g$ into DiTs, but remove the keyframe-LoRA weight matrix $Q_{key}$, keeping the rest unchanged. This variant tests the importance of keyframe-aware LoRA adaptation.

- **w/o $L_{key}^{bg}$**: Use only the background features $L_{bg}$ extracted from $V_{agn}$, without fusing keyframe-based background features, keeping the rest unchanged. This variant validates the role of collaborative background details optimization.

- **w/o Fusion**: Replace weighted fusion of background features with direct concatenation $\text{Concat}(L_m, L_{key}^{max\text{-}bg})$, keeping the rest unchanged. This variant examines the benefit of weighted fusion.

- **w/o CBDO**: We deactivate the entire collaborative background details optimization module, while keeping all other components unchanged. This variant evaluates the contribution of background detail refinement.

- **w/o GDDE**: We deactivate the entire garment dynamic details enhancement module, while keeping all other components unchanged. This variant validates the importance of enriching garment dynamics.

- $F_{key} = 1$: We restrict the keyframe set to only the first frame ($K = 1$) as input, while keeping all other settings unchanged. This variant evaluates the importance of using multiple keyframes.

## H  INSTRUCTION-GUIDED KEYFRAME SAMPLING

---

**Algorithm 1:** Instruction-Guided Keyframe Sampling

---

**Input:** $V_{in}$: Input video (T frames, indices $0 \sim T - 1$ with timestamps $t_0 \sim t_{T-1}$)
Ins: User instruction (*e.g.,* "Show front/back of clothes, raise hand to display sleeves")
Params: $K_{max}$ (max keyframes: 3 for short video, 6 for long video),
$w_1 = 0.3, w_2 = 0.2, w_3 = 0.3, w_4 = 0.2$ (weight coefficients),
$T_{thres}$ (temporal interval threshold: video duration/5),
$Occlu_{thres} = 0.2$ (garment occlusion threshold),
$\lambda = 0.5$ (skeleton difference weight)
**Output:** $F_{key}$: Selected keyframe list (length $\leq K_{max}$)
$View_{targets}, Action_{targets} = parse\_instruction(Ins)$ ;
$F_{anchor} = [\,]$ ;
**for** *each view $\in View_{targets}$* **do**
    $F_{anchor}.append(generate\_standard\_pose(view))$ ;

**for** *each action $\in Action_{targets}$* **do**
    $F_{anchor}.append(generate\_standard\_pose(action))$ ;

$D_{anchor} = [compute\_joint\_direction(f) \mid f \in F_{anchor}]$ ;
$S = [\,]$ ;
**for** $i = 0$ *to* $T - 1$ **do**
    $f = V_{in}[i]$, $t = $ timestamp of $f$ ;
    $S_{ins} = vlm\_score(f, Ins)$ ;
    $D_f = compute\_joint\_direction(f)$ ;
    $S_m = \min(\{cosine\_distance(D_f, d) \mid d \in D_{anchor}\})$ ;
    $cloth\_mask = segment\_garment(f)$ ;
    $S_{cloth} = area(cloth\_mask)/area(f)$ ;
    $occlusion\_ratio = area(occluded\_region(cloth\_mask))/area(cloth\_mask)$ ;
    **if** *occlusion_ratio $> Occlu_{thres}$* **then**
        Continue ;

    $initial\_score = w_1 * S_{ins} + w_2 * (1 - S_m) + w_3 * S_{cloth} + w_4 * 1.0$ ;
    $S.append((i, t, initial\_score))$ ;

$S_{sorted} = sort(S, key = \lambda x : x[2], reverse=True)$ ;
$Idx_{key} = [\,], T_{selected} = [\,]$ ;
**for** *each $(idx, t, score) \in S_{sorted}$* **do**
    **if** $|Idx_{key}| \geq K_{max}$ **then**
        Break ;

    **if** $T_{selected}$ *is empty* **then**
        $S_t = 1.0$ ;
    **else**
        $min\_t\_dist = \min(\{|t - t_s| \mid t_s \in T_{selected}\})$ ;
        $S_t = min\_t\_dist/T_{thres}$ ;

    $final\_score = score * S_t$ ;
    **if** $Idx_{key}$ *is empty* **then**
        $Idx_{key}.append(idx)$ ;
        $T_{selected}.append(t)$ ;
    **else**
        $min\_score\_diff = \min(\{|final\_score - S[ik][2]| \mid ik \in Idx_{key}\})$ ;
        **if** *min_score_diff $\geq 0.1$ and min_t_dist $\geq T_{thres}$* **then**
            $Idx_{key}.append(idx)$ ;
            $T_{selected}.append(t)$ ;

$F_{key} = [V_{in}[idx] \mid idx \in Idx_{key}]$ ;
**return** $F_{key}$ ;

---

## I   MORE QUALITATIVE RESULTS

Fig. A1–Fig. A3 present additional visual comparisons between our KeyTailor and SOTA methods on the ViViD dataset, while Fig. A4–Fig. A6 show additional results on our self-collected ViT-HD. It is evident that KeyTailor produces more realistic and natural videos, capturing finer garment dynamics, preserving coherent background details, and maintaining temporal consistency across frames compared to existing methods.

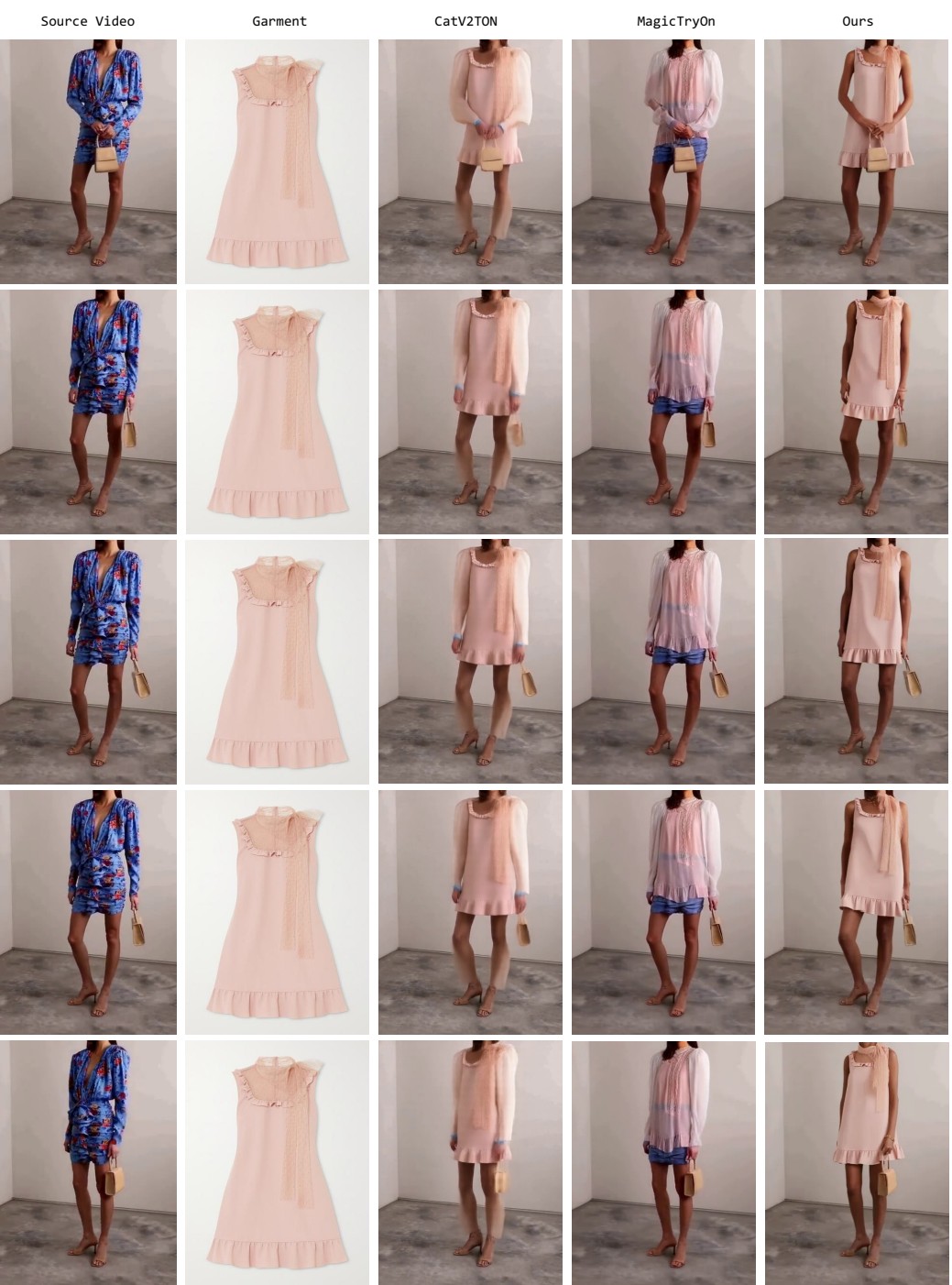

Figure A1: Additional qualitative comparison results on the ViViD dataset. Please zoom in for more details.

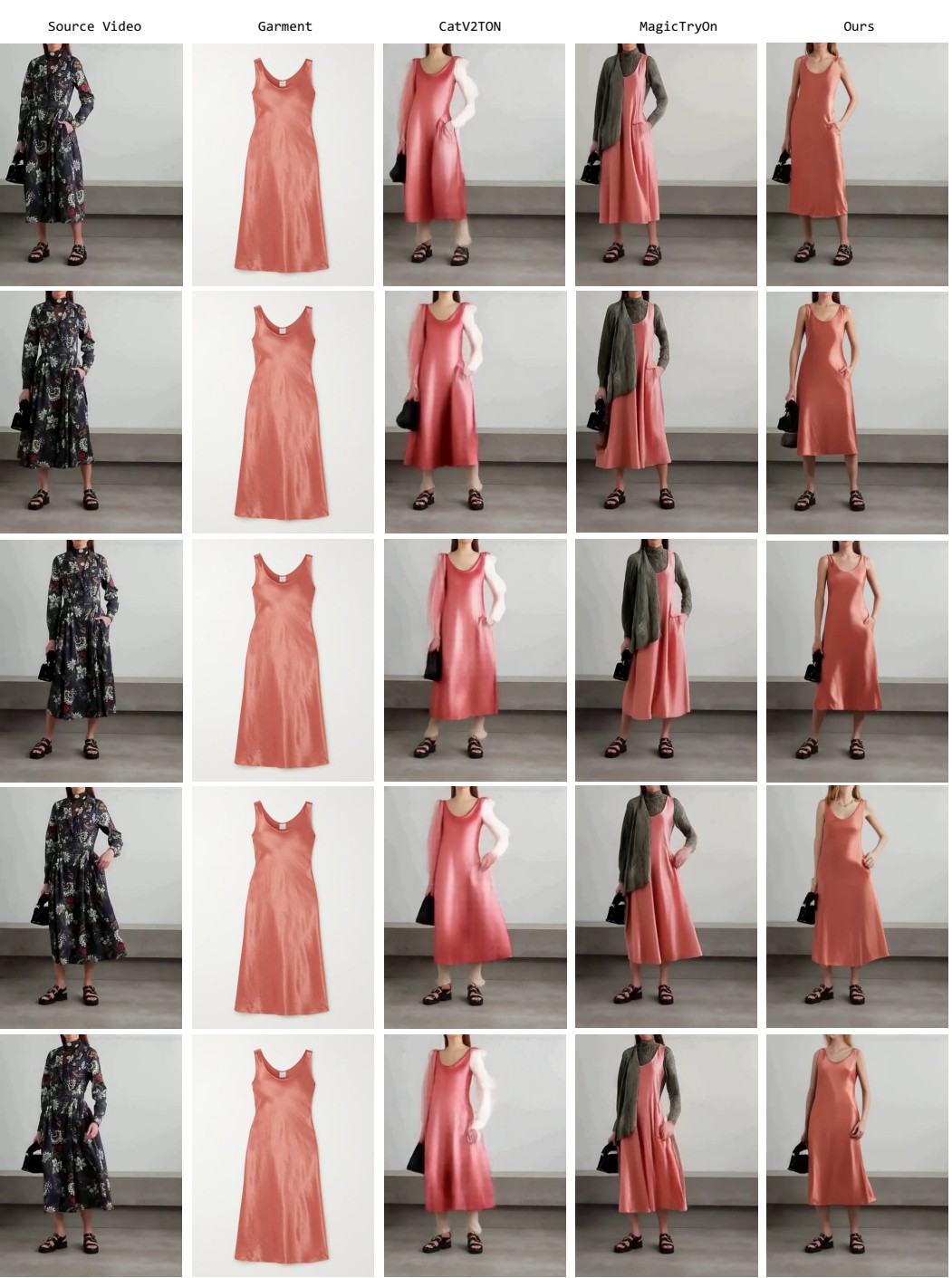

| Source Video | Garment | CatV2TON | MagicTryOn | Ours |

Figure A2: Additional qualitative comparison results on the ViViD dataset. Please zoom in for more details.

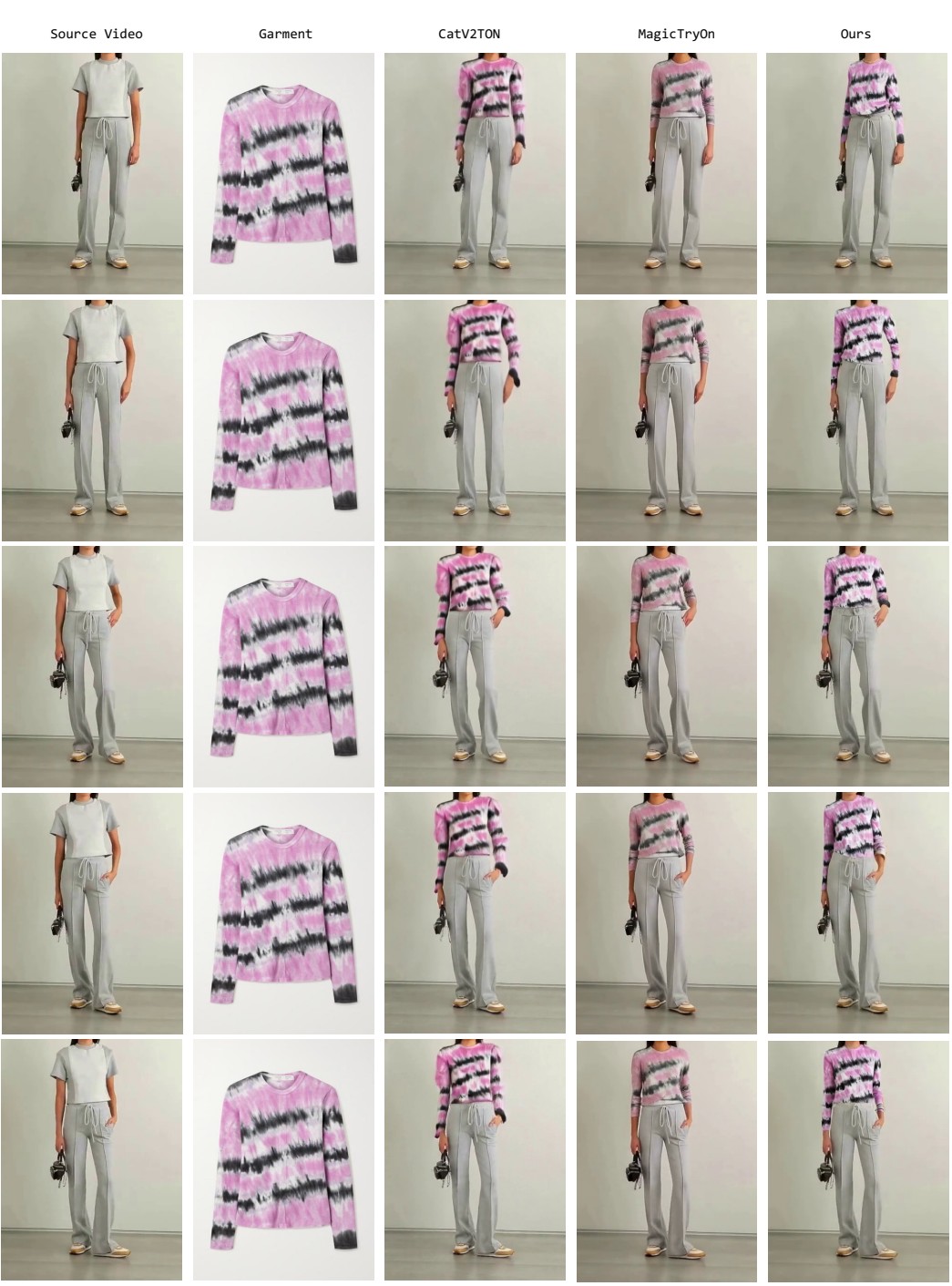

Figure A3: Additional qualitative comparison results on the ViViD dataset. Please zoom in for more details.

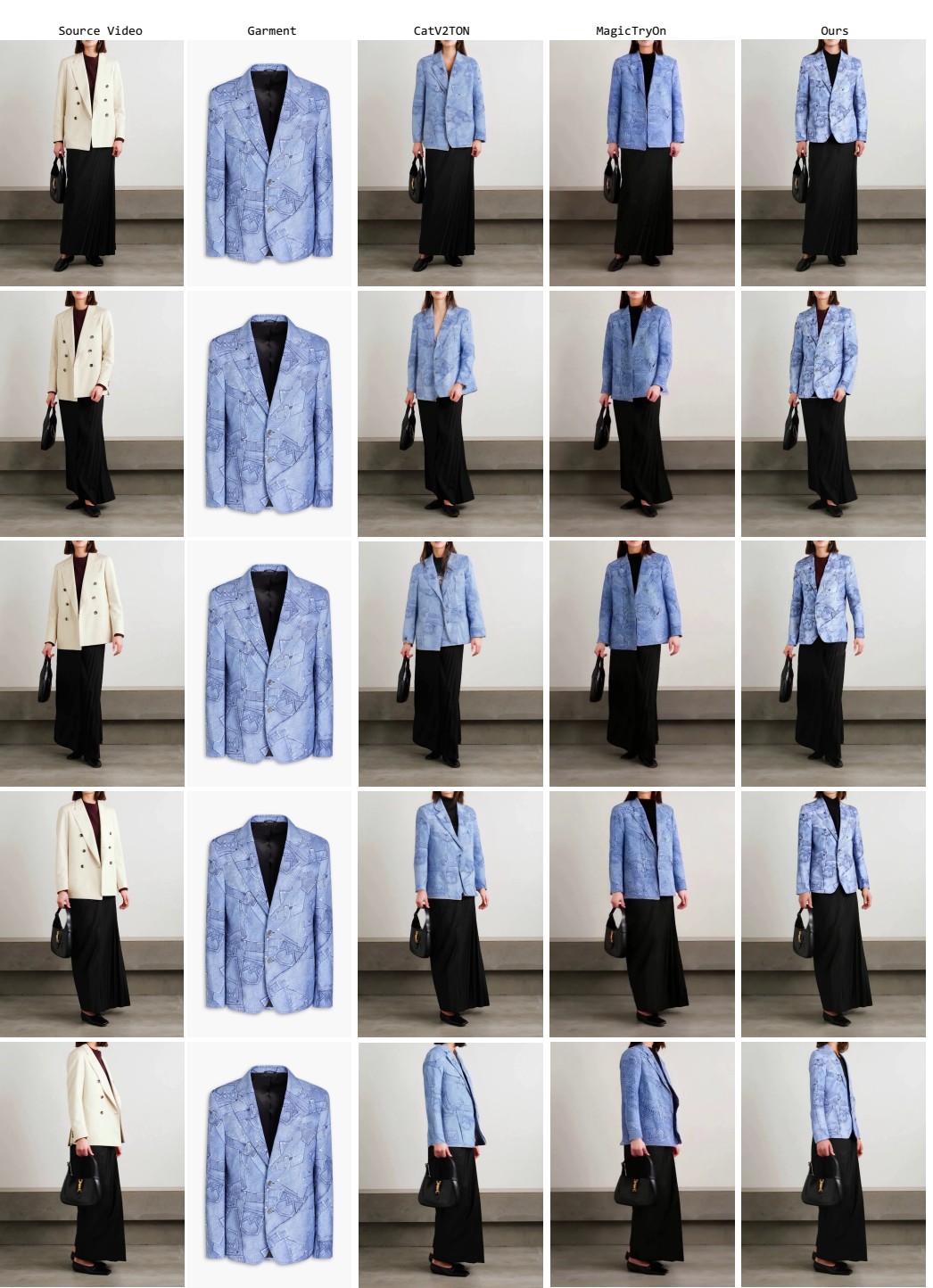

Figure A4: Additional qualitative comparison results on our ViT-HD dataset. Please zoom in for more details.

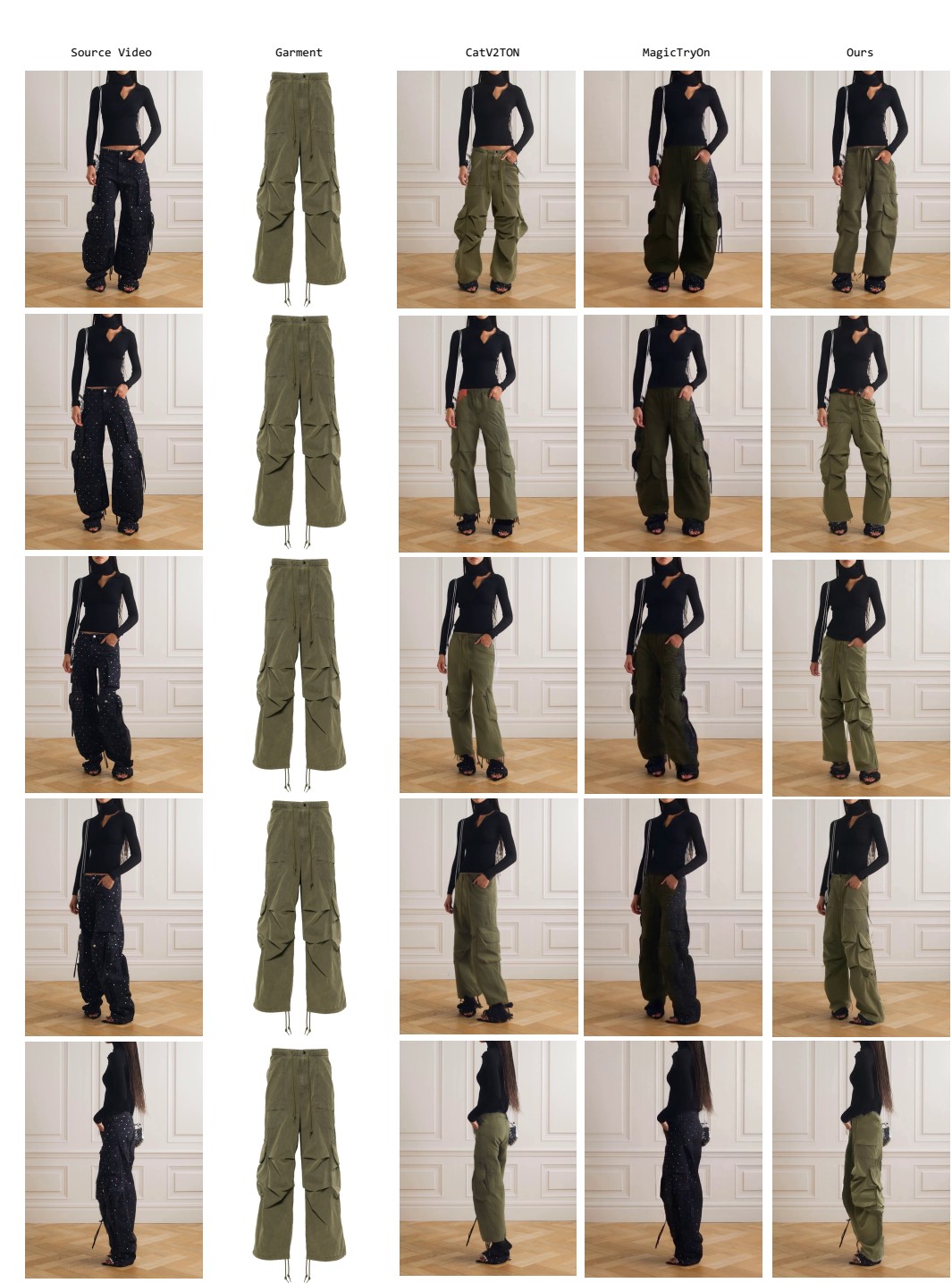

Figure A5: Additional qualitative comparison results on our ViT-HD dataset. Please zoom in for more details.

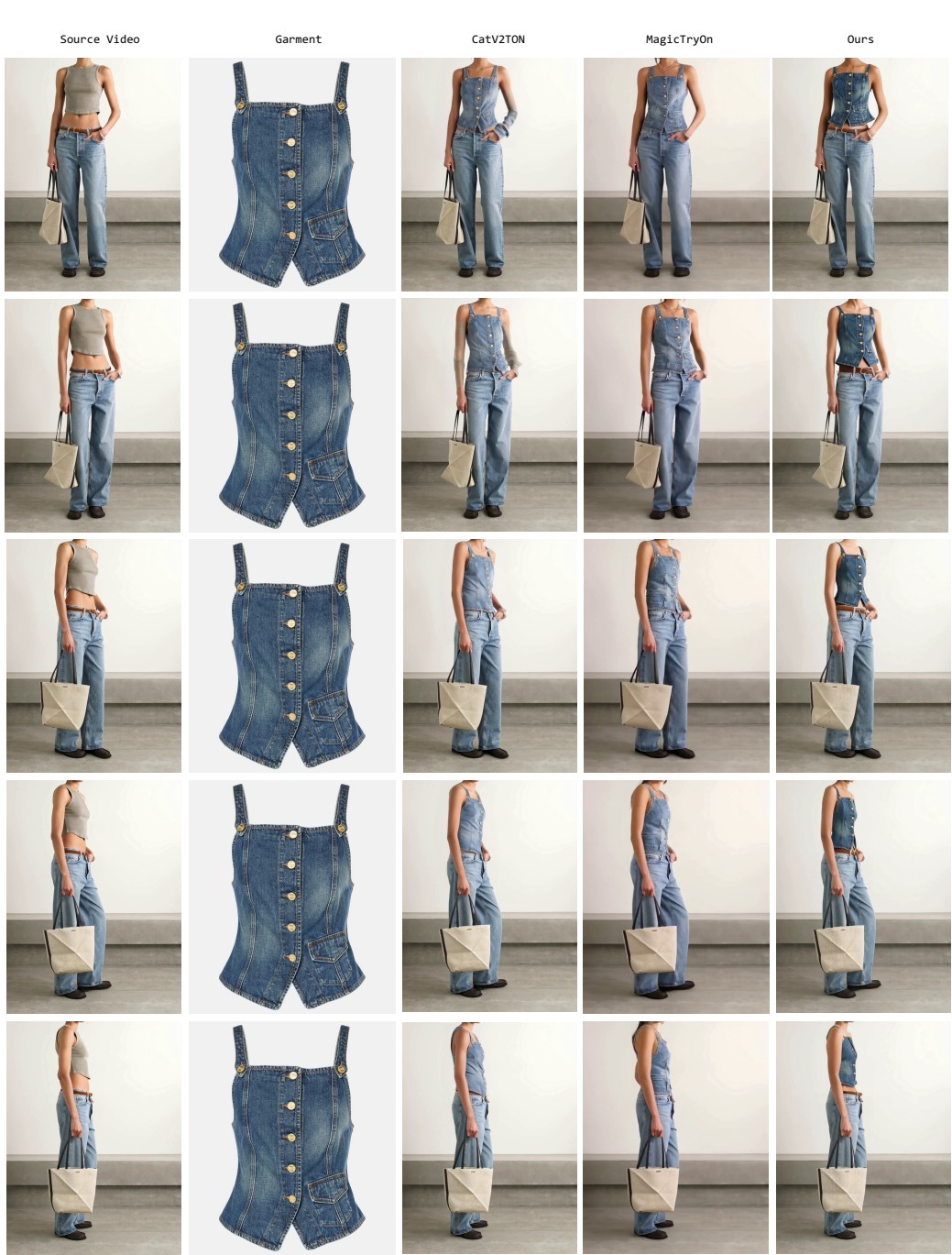

Figure A6: Additional qualitative comparison results on our ViT-HD dataset. Please zoom in for more details.

