# OpenReview forum: "The devil is in the details: Enhancing Video Virtual Try-On via Keyframe-Driven Details Injection"
_ICLR.cc/2026/Conference — ICLR 2026 Conference Withdrawn Submission_

### Official Review · Reviewer_DhR8 · 2025-10-18

**Soundness:** 2
**Presentation:** 2
**Contribution:** 2
**Rating:** 4
**Confidence:** 4

**Summary:**

This paper introduces KeyTailor, a novel framework for video virtual try-on (VVT) that aims to generate high-fidelity and temporally consistent results. The core contribution is a "keyframe-driven details injection" strategy built upon a Diffusion Transformer (DiT) backbone. Instead of heavily modifying the DiT architecture, KeyTailor uses an instruction-guided module (IKS) to sample informative keyframes from the source video. These keyframes are then used by two lightweight modules—the Garment Dynamic Details Enhancement (GDDE) and the Collaborative Background Details Optimization (CBDO)—to enrich garment and background details, respectively. These enriched latents are then injected into the standard DiT blocks. Additionally, the authors introduce a new large-scale dataset for VVT. The paper presents extensive experiments demonstrating that KeyTailor achieves state-of-the-art performance on several benchmarks.

**Strengths:**

*Significant Dataset Contribution*: The introduction of the ViT-HD dataset is a major contribution to the field. Data scarcity, particularly for high-resolution and diverse videos, has been a significant bottleneck. Providing a large-scale (15,070 samples), high-resolution (810 × 1080) dataset will undoubtedly facilitate future research and more robust model development.

*Efficiency and Scalability*: By building upon a standard DiT and using parameter-efficient LoRA for fine-tuning, the proposed method is computationally efficient compared to approaches that require architectural modifications or massive training parameters. This makes the approach more scalable and practical.

**Weaknesses:**

*Crucial Lack of Video-Based Evidence*: The paper's primary claims revolve around enhancing garment dynamics and preserving background integrity across frames—both are temporal properties. However, the evaluation relies almost exclusively on static images (Figs 1, 5, 6, and all appendix figures). Without providing video comparisons for the main results and the ablation studies, it is impossible to verify the claimed improvements in temporal consistency. This is a critical omission, as static frames can be cherry-picked and may hide temporal artifacts like flickering or inconsistency, which the method claims to solve.

*Questionable Ablation Study Design*: The ablation study (Table 5) appears to be designed with an unusually weak baseline, which may exaggerate the importance of each proposed component.
- The "w/o IKS" and "F_key = 1" variants show a catastrophic drop in performance, making them significantly worse than prior SOTA methods like CatV2TON, despite KeyTailor being built on a much stronger backbone (Wan2.1).
- A more convincing ablation would start from a strong baseline (e.g., a DiT conditioned on the flat garment image, following previous paradigms) and add the proposed keyframe-driven modules. The current study does not demonstrate the marginal benefit of the proposed techniques over a competitive baseline; rather, it shows that a crippled version of the model performs poorly.
- It is also suspicious that the removal of nearly every single component leads to a massive performance degradation. A progressive, additive study would provide clearer insight into the individual contribution of each module.

*Over-reliance on Heuristics and Under-Ablated Hyperparameters*: Several key design choices are based on heuristics and hard-coded hyperparameters that are not sufficiently justified.
- IKS Module: The keyframe scoring function Sf(f) depends on a hyperparameter λ, and the process for selecting this value is not discussed. The module also relies on a proprietary VLM (Qwen) for instruction parsing, but its accuracy and robustness are not evaluated.
- CBDO Module: The decision to use only a single keyframe for background detail injection seems limiting, especially for videos with significant camera motion or changing backgrounds. Furthermore, the weighting parameter α is set to a default of 0.3 without any ablation or justification, making it unclear if this value is optimal or dataset-specific.

*Potential for Unfair Comparison*: The paper states that for training, ViT-HD is combined with ViViD. When evaluating on the ViViD benchmark (Table 3, Right), this gives KeyTailor an unfair advantage, as it is trained on a much larger and more diverse dataset (ViT-HD + ViViD) than the baseline methods, which were likely trained only on ViViD. For a direct and fair comparison, KeyTailor should also be trained exclusively on the ViViD dataset when evaluating on its test set. Additionally, the method incorporates a pre-trained single-image try-on model, which introduces a significant number of external parameters and complicates the claim of being a lightweight solution that "avoids additional complexity."

*Lack of Transparency in Data Annotation*: The ViT-HD dataset uses BLIP-2 for automated garment categorization. However, the paper does not include any evaluation of this annotation pipeline's accuracy. Errors in automated labeling could introduce noise into the dataset and affect model training and evaluation, but the extent of this potential issue is not addressed.

*Unclear Data Filtering Methodology*: The paper states that videos with "incomplete clothing occlusion" and "overexposed frames" were discarded during data processing. The specific methodologies, algorithms, or thresholds used to make these determinations are not described. This lack of detail hinders the reproducibility of the dataset curation process.

**Questions:**

- Could the authors provide an anonymous repository with video results? Specifically, videos corresponding to the qualitative comparisons (Fig. 5, Fig. 6) and, most importantly, the ablation study (Fig. 8) are needed to substantiate the claims of improved temporal consistency.
- Regarding the ablation study's baseline: Could you clarify why the "w/o IKS" and "F_key=1" variants perform so poorly, even compared to methods built on weaker backbones? Can you provide an additive ablation study that starts with a strong, standard VVT baseline and shows the incremental improvements from adding the IKS, GDDE, and CBDO modules?
- Regarding the IKS module: How was the balancing hyperparameter λ determined, and how sensitive is the model to its value? What is the estimated accuracy of using Qwen for instruction parsing, and have you considered how errors from the VLM would impact keyframe selection?
- Regarding the CBDO module: How does the single-keyframe approach for background enhancement handle videos with substantial camera movement? Please provide an ablation on the α parameter to demonstrate its impact.
- Regarding the dataset annotation: What steps were taken to validate the accuracy of the BLIP-2 garment annotations? Could you report the quantitative accuracy of the labeling process?
- Regarding the data filtering: Could you please detail the specific criteria and thresholds used to identify and remove frames with incomplete clothing occlusion and overexposure?
- Regarding the experimental setup: For the comparisons on the ViViD benchmark, why not also report results for a version of KeyTailor trained only on the ViViD training set? This would provide a more direct and fair comparison against prior work.

**Details Of Ethics Concerns:**

The paper mentions that the ViT-HD dataset was created by downloading data from "multiple e-commerce platforms." The authors should provide more details on this process to clarify that it was done in compliance with the platforms' terms of service and to address potential copyright issues associated with the video and garment data.

---

### Official Review · Reviewer_4cou · 2025-10-21

**Soundness:** 2
**Presentation:** 3
**Contribution:** 3
**Rating:** 6
**Confidence:** 5

**Summary:**

This paper proposes KeyTailor, a video virtual try-on framework based on diffusion transformers (DiTs), introducing a keyframe-driven details injection strategy to enhance both garment dynamics and background integrity in generated video sequences. Unlike prior works relying on complex interaction modules or modifications to the DiT architecture, KeyTailor achieves state-of-the-art visual fidelity and temporal consistency through two lightweight modules that inject garment and background details from informative keyframes. Additionally, the paper introduces ViT-HD, a large-scale, high-resolution dataset for video virtual try-on, and demonstrates KeyTailor’s superiority over recent baselines through extensive quantitative, qualitative, and user study results.

**Strengths:**

1. The paper significantly advances the ecosystem with ViT-HD, a high-resolution, diverse dataset that bridges a major gap in current VVT research.
2. The proposed keyframe-driven details injection mechanism is well-motivated and architecturally elegant, introducing garment dynamic and background cues, and this design is substantiated by the comparative analysis.
3. Detailed ablation studies and module analysis show the necessity and contribution of each component, supporting the claim that both garment and background detail modules are essential.

**Weaknesses:**

1. I think the masks sequence is not necessary, and this even limit the freedom of generation.
2. The parameters and time consumption superiority seems mainly come from the Wan2.1. The key frame driven injection strategy is well-motivated, but not the main reason.
3. It is recommended to provide more examples in the supplementary materials, such as interchange pairings of skirts and trousers or shirts and outerwear.

**Questions:**

See weakness.

---

### Official Review · Reviewer_3EuG · 2025-10-30

**Soundness:** 2
**Presentation:** 3
**Contribution:** 2
**Rating:** 2
**Confidence:** 5

**Summary:**

This paper proposes KeyTailor, a DiT-based video virtual try-on (VVT) framework leveraging keyframe-driven detail injection. The authors propose two key innovations: (1) a lightweight framework that enhances garment fidelity and background integrity without modifying the DiT architecture, and (2) ViT-HD, a large-scale high-definition dataset containing 15,070 samples.

**Strengths:**

1. The keyframe-driven detail injection strategy enhances performance without modifying the DiT architecture, achieving a lightweight design that balances effectiveness and computational efficiency.

**Weaknesses:**

1. While the keyframe-driven approach is novel, the underlying technical components rely heavily on existing techniques: DiT backbone from Wan2.1, image-based try-on model, standard VAE encoders, and conventional attention mechanisms. The paper lacks fundamental architectural innovations that would push the boundaries of generative modeling for virtual try-on.
2. The instruction-guided keyframe sampling module lacks rigorous ablation studies on the impact of different instruction types, keyframe quantities (beyond K=1 vs multiple), and selection criteria. The paper doesn't explore how sensitive the model performance is to suboptimal keyframe selection in real-world scenarios.
3. The paper presents impressive results but lacks analysis of failure modes, edge cases, or scenarios where the keyframe-driven approach might fail. Important questions remain unanswered: How does the method handle rapid motion sequences? What happens when keyframes contain motion blur? How robust is the method to inaccurate pose estimations?

**Questions:**

See above

---

### Official Review · Reviewer_Pcqg · 2025-10-31

**Soundness:** 3
**Presentation:** 4
**Contribution:** 3
**Rating:** 6
**Confidence:** 4

**Summary:**

This paper aims to address the limitations of existing Diffusion Transformer (DiT)-based video virtual try-on (VVT) methods in capturing fine-grained garment dynamics and maintaining background consistency. It also notes the high computational costs and dataset limitations of these methods. To this end, the authors propose a novel framework, KeyTailor, which features a "keyframe-driven details injection strategy" at its core. KeyTailor uses an instruction-guided module to sample informative keyframes. It then extracts garment and background details from these keyframes via two specialized modules (GDDE and CBDO), injecting these details into the standard DiT backbone. Furthermore, the paper contributes a large-scale, high-definition video dataset, ViT-HD, comprising 15,070 samples at  resolution. Experiments demonstrate that KeyTailor, without modifying the DiT architecture, outperforms existing SOTA methods in terms of garment fidelity and background integrity.

**Strengths:**

* The core contribution is the novel and efficient KeyTailor framework. Its "keyframe-driven details injection" strategy (implemented via the GDDE and CBDO modules) is an intelligent design. It enriches details by enhancing input latents rather than adding complex interaction layers within the DiT backbone, which stands in contrast to prior work.

* The method offers significant computational efficiency. As KeyTailor is built on a standard DiT and fine-tuned only with LoRA, it avoids the massive parameters and high training costs associated with other SOTA DiT methods that introduce extra modules (as shown in Fig. 2(b) and 2(c)).

* The paper's second major contribution is the ViT-HD dataset. This large-scale (15,070 samples), high-resolution ($810 \times 1080$) VVT dataset addresses the long-standing bottleneck of data scarcity and low quality in the field, providing significant value to the community.

* The method demonstrates state-of-the-art performance in quantitative (Tables 2, 3) and qualitative (Figs. 1, 5, 6) experiments across multiple datasets, including ViT-HD, VVT, and ViViD. The improvements are particularly evident in preserving garment details (like wrinkles) and background consistency.

* The paper provides thorough ablation studies (Table 5) that clearly validate the effectiveness of each new component (e.g., IKS, GDDE, CBDO). Furthermore, the user study (Fig. 7) confirms the superior perceptual quality of the generated results.

**Weaknesses:**

* The reliance of the keyframe sampling (IKS) module on an external large vision-language model (Qwen) and predefined instructions may be a weakness. How robust is this strategy if the video content does not match the instruction (e.g., "Show the front and back of the garment")?

* The generalizability of the IKS instructions is questionable. The instruction used seems well-suited for e-commerce data but not for "in-the-wild" scenarios (e.g., sports, movie clips) where people do not perform such specific actions. It is unclear how the strategy would perform in these more general contexts.

* The data preprocessing pipeline (Section 2) and the model's inference flow (Fig. 4) are quite complex, requiring multiple components like OpenPose, HumanParsing, and image inpainting to generate the "agnostic" video. This could be a barrier to practical application.

* The Garment Dynamic Details Enhancement (GDDE) module relies heavily on a pre-trained single-image try-on model applied to the first frame. If this initial frame is of poor quality, occluded, or at an odd angle, this error could be amplified and propagated throughout the video sequence.

* While the paper shows many successful cases, it lacks a thorough analysis of failure cases. Under what conditions does KeyTailor fail? For example, with extremely complex motion, very loose clothing (like a cape), or highly dynamic backgrounds.

* The efficiency comparison is not intuitive enough. While Fig. 2(c) shows FLOPs and time per iteration, a more direct end-to-end inference speed comparison (e.g., time to generate one second of video) against MagicTryOn and CatV2TON is missing.

**Questions:**

* Regarding the IKS module: Could the authors elaborate on the robustness of the instruction-guided keyframe sampling (IKS) module? If the input video is a simple "in-the-wild" clip (e.g., a person walking down the street) that does not contain the actions specified in the instruction, how does the sampling proceed? Does it fail or degrade gracefully to a reasonable sampling?

* Regarding the GDDE module: The GDDE module uses a single-image try-on model on the first "agnostic" frame. Have the authors tried using other frames (e.g., the "best" keyframe selected by IKS) as the base for the single-image try-on? How sensitive is the model's performance to the quality of this initial frame?

* Regarding the ViT-HD dataset: ViT-HD is a great contribution. The authors mention discarding videos with "incomplete clothing occlusion". Can this be quantified? What percentage of the originally collected videos were discarded? This would help in understanding the "in-the-wild" nature of the raw data.

* Regarding the CBDO module: In the Collaborative Background Details Optimization (CBDO) module, a balance weight is used to fuse global background features and keyframe background features. How sensitive is the model to this hyperparameter? Is this value fixed for all cases?

* Regarding limitations: Could the authors discuss the main limitations of the model? For instance, how does KeyTailor handle garments with significant topological changes (e.g., zipping up a jacket, putting on a hood)?

---

### Note · Authors · 2025-11-14

I have read and agree with the venue's withdrawal policy on behalf of myself and my co-authors.